# Phytosterol-Enriched Dietary Supplements for Lowering Plasma LDL-Cholesterol: Yes or No?

**DOI:** 10.3390/nu17040654

**Published:** 2025-02-12

**Authors:** Frans Stellaard, Dieter Lütjohann

**Affiliations:** Institute of Clinical Chemistry and Clinical Pharmacology, University Hospital Bonn, 53127 Bonn, Germany; fstellaard@hotmail.com

**Keywords:** phytosterols, atherosclerosis, cardiovascular disease, campesterol, sitosterol, cholesterol

## Abstract

Elevated plasma low-density lipoprotein cholesterol (LDL-C) is associated with an increased risk for atherosclerosis and development of cardiovascular disease. An elevated plasma LDL-C concentration is the result of enhanced C synthesis, C absorption, and/or altered C homeostasis. Plasma LDL-C lowering can be achieved using pharmaceutical means. Statin therapy inhibits endogenous C synthesis and leads to a mean 40% LDL-C reduction. Ezetimibe inhibits C absorption and achieves an average 20% LDL-C reduction with a 10 mg daily intake. Phytosterol therapy is established by dietary supplements enriched in phytosterols and/or phytostanols. A dosage of 2 to 3 g a day reduces C absorption and leads to an average 10% LDL-C reduction. This dosage expresses a 10-fold increased daily intake for phytosterols or a 100-fold increased intake of phytostanols. Phytosterol- and -stanol-enriched dietary supplements are freely available in the supermarket. The majority of consumers may be healthy subjects with a plasma LDL-C in the normal range. Scientific evidence reveals that increased phytosterol intake may be associated with the development of atherosclerosis. The degree of increased risk is dependent on the patient’s genetic polymorphisms in NPC1L1 and ABCG5/G8 transport proteins as well as on the established risk reduction due to LDL-C lowering. Subjects with a normal or only slightly elevated LDL-C have only minimal LDL-C lowering and lack the compensation for the potential increased risk for atherosclerosis by phytosterols.

## 1. Introduction

It is well recognized that an elevated plasma cholesterol (C) is associated with an increased risk to develop atherosclerosis and, consequently, cardiovascular heart disease (CHD) [1,2]. The plasma low-density lipoprotein C (LDL-C) concentration is considered the most atherogenic, although the triglyceride (TG)-richer very low density C (VLDL-C) and intermediate density C (IDL-C) and chylomicron and chylomicron remnant C have been shown to exhibit atherogenic properties as well [3,4,5]. However, these are generally much lower in concentration. Plasma LDL-C lowering is initiated to decrease the risk for CVD development. It has been estimated that an LDL-C reduction of 1 mmol/L induces a 22% risk reduction for CHD [6]. The input fluxes of C into the body C pool are endogenous C synthesis and C absorption. The flux of C absorption consists of dietary C and endogenous C entering the intestine via biliary secretion and direct bile-independent transintestinal excretion (TICE) [7]. The fractional C absorption rate varies from 20 to 80% [8]. Plasma LDL-C lowering is initially based upon reduction in one of the two or both C input fluxes. Pharmacological treatment is strongly advised in the responding guidelines for the treatment of hypercholesteromia, starting with statin treatment, which reduces endogenous C synthesis and increases LDL-receptor (LDL-R) activity [9,10]. The first described statin shown to inhibit cholesterologenesis was compactin [11]. Statin treatment encounters two problems. The first is that many patients experience serious side effects and an extreme statin intolerance. Intolerance to a statin will be followed by the choice of another statin. The second is that the efficiency in terms of LDL-C lowering is highly variable, scoring between 10 and 60% [12]. Based on the baseline plasma LDL-C value and additional clinical risk factors like age, inactivity, high blood pressure, obesity, diabetes, and being male, a target is set for the LDL-lowering to be reached in the specific patient. It has been suggested that two types of patients may be distinguished, high C synthesizers reacting well on statin treatment and high C absorbers that require reduction in C absorption [13]. The pharmacological approach led to the development of ezetimibe, which reduces the activity of the transport protein Niemann–Pick C1-Like 1 (NPC1L1) that enables or reduces uptake of C from micelles in the intestinal lumen into the enterocyte. NPC1L1 is also present in the liver and leads to reabsorption of C from bile into the hepatocyte [14]. Applying a dose of 10 mg/day ezetimibe, an average 20% reduction in plasma LDL-C is obtained [12]. In case the high-dose statin treatment is not well tolerated or in case the LDL-C lowering by a statin does not reach the intended goal for LDL lowering, the statin dose can be lowered, and treatment is combined with ezetimibe [9,15]. The neutraceutical approach is diverse. Many food products, like ω3- and ω6-rich ones like fish but also olive oil, garlic, and many other products [16], are promoted to reduce plasma C. However, this requires a complete adaptation to a different diet. A vegan diet is low in dietary C intake and can create a significantly lower plasma LDL-C [17]. As an alternative, dietary supplements enriched in phytosterols and phytostanols have been developed. These are based on the knowledge that these sterols competitively reduce the uptake of C into intestinal micelles transporting fats and sterols through the intestine [18]. This leads to a reduction in C absorption and, on average, a 10% reduction in plasma LDL-C [19]. It must be emphasized that a reduction in C absorption by ezetimibe or phytosterols induces an increase in C synthesis [20]. This counteracts the effect of C absorption reduction on LDL-C lowering. A combination of statin treatment with ezetimibe or phytosterol treatment enhances the efficacy of mono treatments to an average of 60% [12].

The specific biological function of phytosterols is unknown. Physiological beneficial associations have been indicated in relation to gastrointestinal tracts, including anti-inflammatory and hepatoprotective activity as well as the anti-cancer properties and the impact on the gut microbiome [21,22,23].

In this review, we discuss the mechanisms of action of the pharmacological and neutraceutical approaches to lower C absorption. We selected publications based on the general topic combination of C and phytosterols in PubMed and selected searches based on general findings. Starting in 1953, O.J. Pollak showed, for the first time, C reduction in men by plant sterols [24]. In 1957, Cytellin^®^ was introduced by Eli Lilly in Indianapolis, IN, USA (6–18 g/day) and widely used in the pre-statin era with unpleasant side-effects in the gastro-intestinal tract. Results were rather variable. It was never widely used and stopped in 1982, briefly before statins came onto the market [25]. After a first study showing C reduction in men by plant stanols [26], a genius idea from a Finnish scientist, Dr. Ingmar Wester, led to the introduction of esterified plant stanols added to plant-based margarine “Benecol” [27], followed by the introduction of “pro Active” margarines, fortified with plant sterol esters in the United States of America, Australia (1999), and in the European Union in 2000 [25]. Also, the differences in treatment efficiencies and the risks introduced by the treatment strategies will be compared. The general term phytosterols combines the unsaturated phyosterols and its 5α-saturated phytostanols (Figure 1). When necessary for understanding, phytostanols will be distinguished from phytosterols and mentioned as such.

## 2. Cholesterol Absorption Related to Cardiovascular Disease

Various publications present evidence that enhanced C absorption is a specific risk factor for cardiovascular disease, more than enhanced C synthesis [28,29,30]. Direct measurement of C absorption using stable isotope technology [8] is a time-consuming, expensive technique requiring a three day feces collection, which is not suitable for large-scale investigations. Marker technology has been developed to express the status of a patient’s C synthesis and C absorption. Although more markers have been described, the plasma campesterol/C ratio is generally used as the marker for C absorption and the lathosterol/C ratio as marker for C synthesis [31]. Since enhanced C absorption is generally associated with a reduced C synthesis, the plasma campesterol/lathosterol ratio is considered the best marker to express enhanced C absorption and to define patients as high absorbers [32,33]. Using this marker, it has been documented that a high campesterol/lathosterol ratio or a low lathosterol/campesterol ratio is associated with cardiovascular disease. Weingärtner et al. [34] showed that the campesterol/lathosterol ratio exerted an odds ratio of 3.3 with a statistical significance, given by a *p*-value of 0.016, as a high cardiovascular risk factor. Nasu et al. presented evidence that the plasma campesterol/lathosterol ratio is significantly correlated with the plaque vulnerability [35]. The general interpretation of these data is that absorbed C is more atherogenic and prone to develop cardiovascular disease. Campesterol is a phytosterol with a relatively high fractional rate of absorption. The mechanisms of phytosterol absorption and C absorption are the same, although at different levels. An enhanced absorption of C is accompanied by an enhanced absorption of campesterol. That is the reason why campesterol is considered a good marker for C absorption. The concentration in plasma is corrected for the C concentration to correct for lipoprotein metabolism. Both concentrations of C and campesterol in plasma increase when sterol absorption increases. Enhanced C absorption is partly compensated by reduced C synthesis. Enhanced campesterol absorption is not compensated. Thus, the plasma campesterol/C ratio does not reflect the ratio of absorbed campesterol and absorbed C. The interpretation of the association between the plasma campesterol/lathosterol ratio and the risk for cardiovascular disease is complex. An increased ratio reflects enhanced C absorption. However, it also reflects enhanced campesterol absorption and potentially also the absorption of phytosterols in general. What factor is the causal factor for the development of atherosclerosis?

## 3. Absorption of Cholesterol and Phytosterol Absorption

C is present in food constituents from animal origin and present in the lipid fraction in the free and esterified form. The daily intake is on average about 300 mg/day and may vary from 200 to 500 mg/day. After passage through the stomach, C-ester is de-esterified by pancreatic cholesterol esterase supplied by the pancreatic juice secreted into the upper small intestine. Free C is insoluble in water and is incorporated in micelles formed by bile acids supplied by the bile flow through activation of gallbladder contraction. Together with the bile acids, biliary C enters the intestine and is taken up into the micelles. Within the micelles, C is transported down-stream the small intestine. When the micelles enter the unstirred water layer along the enterocytes, C is released and partly transported into the enterocytes. This transport is created by the sterol transport protein Niemann–Pick C1-Like 1 (NPC1L1) [36,37]. Within the enterocyte re-esterification takes place and C-ester is incorporated into chylomicrons and secreted into blood via the lymph. The non-esterified fraction is re-secreted into the intestinal lumen by the dimeric ATP-binding cassette sub-family G member 5 and 8 (ABCG5/G8) transport protein [38]. Measured over a three-day period, on average, 50% of C is absorbed, varying from 20 to 80% [8] (Figure 2).

Phytosterols consisting of plant sterols and stanols are present in plant-derived food products [39]. Many different sterols and also stanols are known, of which campesterol, brassicasterol, sitosterol, and campesterol are the most abundant sterols, and 5α-campestanol and 5α-sitostanol are the most abundant 5α-stanols. 5α-campestanol is the saturated form of campesterol and brassicasterol, and 5α-sitostanol is the saturated form of sitosterol and stigmasterol (Figure 1). The sterols are unsaturated, having a double bond between C4 and C5 positions in the steroid nucleus, like C has. Stanols are 5α-saturated after addition of hydrogen atoms, missing the double bond. Compared to C, phytosterols contain alkyl groups in the side chain. They are also incorporated into the intestinal micelles and with a higher efficiency than C. Their uptake into and metabolism inside the enterocytes are quantitatively extremely different from C and also different between sterols and stanols. On average, the absorption efficiencies of sitosterol and campesterol are 0.5% and 2%, respectively, in healthy subjects. The values for sitostanol and campestanol are 0.04 and 0.2% [40]. These results were obtained with stable isotope technology applying serum measurements. Lütjohann et al. applied continuous stable isotope-feeding technology and fecal sampling and found 5% for sitosterol and 16% for campesterol for the fractional C absorption rate [41]. The low absorption rates for phytosterols are caused by a low interaction with NPC1L1, low esterification rate in the enterocyte, and high interaction with intestinal ABCG5/G8. Importantly, NPC1L1 and ABCG5/G8 are also present in the liver, regulating the biliary C secretion. NPC1L1 initiates partial reabsorption of C from the hepatic bile duct, and ABCG5/G8 promotes biliary C secretion. The low rate of interaction between phytosterols and hepatic NPC1L1 and high rate of interaction between phytosterols and hepatic ABCG5G8 causes the small amount of absorbed phytosterols to be effectively eliminated via biliary secretion.

## 4. Plasma Low-Density Lipoprotein Cholesterol Lowering

### 4.1. Pharmaceutical Therapy

Patients with elevated plasma LDL-C concentrations must be treated according to the guidelines of the American Heart Association [42] and the European Society of Cardiology [43]. Patients diagnosed with atherosclerotic cardiovascular disease (ASCVD) must be treated accordingly to prevent development of coronary heart disease (CHD). Patients that have already experienced CHD must be treated to prevent a second event [44]. A risk assessment must be made based on experienced coronary events (heart attack, stroke), plasma LDL-C level, and additional risk factors, such as age, inactivity, high blood pressure, obesity, diabetes, and being male. Patients having experienced cardiovascular events must be treated maximally in order to achieve a ≥50% LDL-C reduction or reach a certain plasma LDL-C concentration like <70 mg/dL for patients at high risk. This requires maximally tolerated statin regime, eventually combined with ezetimibe. When the combined treatment efficiency is still insufficient, additional treatment with a Proprotein convertase subtilisin/kexin type 9 (PCSK9) inhibitor is added [45]. This reduces the catabolism of the LDL-R and improves the removal of LDL particles from blood. Patients who have not yet experienced any cardiovascular events but suffer from many external risk factors as mentioned before, including elevated plasma LDL-C, are treated with a broad strategy, including establishment of a healthier lifestyle (food management, activity management), treatment of obesity, diabetes, and high blood pressure, combined with preventive reduction of plasma LDL-C. Depending on the LDL-C concentration a moderate statin treatment with or without ezetimibe may be subscribed. A patient dependent goal for LDL-C lowering must be set and treatment adjusted to reach the goal. In all strategies statins are first choice. When strong side effects are encountered, statin may be replaced by bempedoic acid [46]. At relatively low risk, the lowest dose of statin may be tested as well as monotherapy with ezetimibe or a combination of both. It has been advised to test the patients status of C synthesis and C absorption. A high synthesizer should be treated with statin or bempedoic acid and a high absorber with ezetimibe. However, Stellaard et al. recently showed that, in healthy subjects, there is no association between C synthesis or C absorption and the plasma LDL-C concentration [47]. Thereafter, Lütjohann and Stellaard studied subjects with mildly elevated plasma LDL-C on treatment with simvastatin, ezetimibe, and combination therapy [48]. They could show that the degrees of LDL-C reduction achieved with all three therapies were not determined by the subject’s baseline C synthesis nor by the baseline C absorption. Moreover, the degree of LDL-C reduction was also not determined by the obtained reduction in C synthesis or C absorption. The baseline plasma LDL-C concentration was the important determinant of the degree of LDL-C reduction.

### 4.2. Phytosterol Application

Any adjustment of food composition leading to a lower plasma LDL-C concentration may be considered a nutraceutical treatment. This review focuses on particular food products fortified with phytosterol and/or -stanol that are being sold in local supermarkets and are available for anyone interested in buying the product. Pollak used unesterified (free) sitosterol to lower total C for the first time in 1953 [24]. Consuming between 5 and 10 g/d, the mean reduction in total C was 25%. In 1995, Miettinen et al. reported a 14% decrease in LDL-C in hypercholesterolemic men consuming 1.8 to 2.6 g/day of sitostanol ester [27]. For the first time, esterified stanols, mainly 5α-sitostanol, were incorporated into a food product, i.e., a margarine. The first commercial food products enriched in plant sterol esters were produced by Unilever (Becel Pro active). Examples of plant sterol ester products include Promise Activ in the United States (Unilever, Englewood Cliffs, NI, USA) and other products, such as CardioAid^TM^ S and CardioAid^TM^ SWD (ADM, Chicago, IL, USA), Vegapure^®^ (BASF, Ludwigshafen, Germany), Corowise^TM^ Plant Sterols (Cargill Inc., Minneapolis, MN, USA), and Danacol^®^ (Danone, Paris, France) [49]. After decades of experience, it can now be stated that phytosterols have similar efficacies independent of the format of the product being sterols in free or conjugated form, unsaturated or saturated form (stanols) and with fat or non-fat carriers, in capsules, tablets, and foods [50]. Phytosterol consumption at dosages of 1.1, 2.1, and 3.3 g/d resulted in plasma LDL-C reductions of 6, 8, and 12%, respectively [19]. Daily intake of phytosterols in healthy subjects consuming a Western diet is up to about 300 mg/day, which is comparable with daily C intake. Vegans may have a daily input up to 600 mg/d. The majority of consumed phytosterols consists of sitosterol, campesterol, and stigmasterol. Daily consumption of phytostanols is much lower and estimated to be 18 to 24 mg/day, consisting of mainly 5α-sitostanol and 5α-campestanol [51]. Although present in all plant material, vegetable oils, nuts, breads, cereals, and vegetables contain mostly phytosterols, cereals most phytostanols. The sterols are extracted from plant material, esterified, concentrated, and incorporated into dietary supplements or in capsules. Dosages of ≥2 g daily are recommended both for sterols and stanols, being more than 10 times the normal intake for sterols and 100 times the normal intake for stanols. The action of phytosterols and -stanols to lower plasma LDL-C is at the level of incorporation of dietary and biliary C into the intestinal micelles, i.e., before absorption [52]. They compete successfully with C for the incorporation into the micelles. The ratio of phytosterol and phytostanol to C increase roughly from 1 to 10 and from 0.1 to 10, respectively, when consuming 2 to 3 g supplement daily. At the moment of consumption of the fortified food or capsule, the ratio may be different and dependent on the C intake at that moment. Is the ≥2 g/day consumed in one portion or divided over more portions per day with a meal or in between meals? These choices determine the suppression of C input into the micelles and the efficacy of treatment. Assuming that the higher intake of phytosterols and -stanols does not lead to alteration of their fractional absorption rate, their daily absorbed amounts (mg/d) also increase 10-fold and 100-fold. The consequences of these increases must be carefully investigated.

## 5. Disadvantages and Side Effects of Treatment

### 5.1. Pharmaceutical Treatment

Lowering of plasma LDL-C by statin treatment via reduction in C synthesis encounters primarily the well-known problem of muscle pain [53]. When the patient is extremely sensitive to even moderate doses of various statins, they are declared statin-intolerable. They may then be treated with bempedoic acid [46]. For treatment based upon reduction in absorption, today, only ezetimibe is available. Today, a standard dose of 10 mg/d is ingested once daily. This dose is generally well tolerated, although many common but tolerable side effects may occur in less than 4% of the patients [54,55].

### 5.2. Phytosterol Treatment

It has been documented that treatment with phytosterols leads to elevated plasma levels of these compounds [56,57]. It also has been documented that high plasma levels are associated with increased risk for atherosclerosis [56,58,59]. The most extreme clinical situation is created by the disease sitosterolemia or better expressed as phytosterolemia caused by genetic alterations of the ABCG5 or ABCG8 genes, leading to the combined heterodimer ABCG5/G5 transport protein. The re-secretion of phytosterols from the enterocyte back into the intestinal lumen and their biliary secretion are highly reduced, and they accumulate in blood and liver [21,60]. ABCG5/G8 mutations also affect C. The intestinal ABCG5/G-dependent re-secretion of C in the healthy state is only limited, but C absorption is still partly enhanced under phytosterolemia. The minimalized ABCG5/G8 activity in the liver and the lower biliary C secretion rate more dominantly lead to enhanced plasma C levels and development of atherosclerosis. However, atherosclerosis is not always a consequence of the disease [61]. Development of atherosclerosis while lowering plasma LDL-C applying phytosterols is under discussion. Different aspects have been published, supporting the increased risk for atherosclerosis development, down-scaling the risk and excluding the risk. Supporting mechanistic evidence is provided by Weingärtner et al. [58]. The authors combined human clinical data with experimental data obtained in wildtype mice and humanized apolipoprotein E knock out mice on a Western-type diet and normal chow diet. The treated mice received a diet with 2% plant sterol esters consisting of mainly phytosterols and at low level of phytostanols for 4 weeks. The phytosterol plasma concentration strongly correlated with increased atherosclerotic lesion formation. Eighty-two patients underwent elective aortic valve replacement, owing to severe aortic stenosis. Aortic cusps were removed from aortic rings. Ten patients had consumed a sterol-ester-supplemented margarine (Becel pro-activ [Unilever Deutschland GmbH, Werk Pratau, Germany]) for more than 2 years before aortic valve replacement. Four of them reported an irregular consumption, averaging one serving/day (0.75 g/d). Six patients consumed at least two servings/day (1.5 g/d) for up to 4 years. The patients with highest intake had a 3-fold higher plasma level of campesterol and a 6-fold higher level of campesterol in the aorta valve cusps. Generally, the campesterol/C ratio in aorta valve cusps was positively correlated with the campesterol/C ratio in plasma. The authors concluded that their findings underline the need for prospective clinical studies with cardiovascular end points for functional foods supplemented with phytosterols. However, until today, no long-term studies with cardiovascular end points have been performed. A genome-wide association study for serum phytosterols was conducted in a population-based sample from KORA (cooperative Research in the Region of Augsburg (*n* = 1495) with subsequent replication in two additional samples (*n* = 1157 and *n* = 1760). The authors concluded that common variants in ABCG5 and at the blood group AB0 are strongly associated with serum phytosterol levels and show concordant and previously unknown associations with CAD [62]. Alleles of ABCG8 and ABO associated with elevated phytosterol levels displayed significant associations with increased CAD risk (single-nucleotide polymorphism [SNP]: rs4245791, odds ratio, 1.10; 95% CI, 1.06 to 1.14; *p* = 2.2 × 10^−6^; SNP rs657152, odds ratio, 1.13; 95% CI, 1.07 to 1.19; *p* = 9.4 × 10^−6^). Hypothesis-supporting data are also described by Helgadottir et al. [63]. The authors examined the effects of ABCG5/8 variants on non-high-density lipoprotein (non-HDL) C (*n* = 610,532) and phytosterol levels (*n* = 3039) and the risk of CAD. From the results, the authors concluded that genetic variation in C absorption affects levels of circulating non-HDL-C and risk of CAD and that both dietary C and phytosterols contribute directly to atherogenesis. A genetic score of ABCG5/8 variants predicting 1 mmol/L increase in non-HDL-C is associated with a 2-fold increase in CAD risk [odds ratio (OR) = 2.01, 95% confidence interval (CI) 1.75–2.31, *p* = 9.8 × 10^−23^] compared with a 54% increase in CAD risk (OR = 1.54, 95% CI 1.49–1.59, *p* = 1.1 × 10^−154^) associated with a score of other non-HD-C variants predicting the same increase in non-HDL-C (*p* for difference in effects = 2.4 × 10^−4^). Contradictory data were obtained by Wilund et al. [64]. Sitosterolemic mice expressed 20-fold increased plasma phytosterol levels due to inactivation of ABCG5/G8. However, aortic lesions were not enlarged in the sitosterolemic mice compared with littermates. The authors also investigated the plasma levels of C and phytosterols in 2542 human subjects and related those with coronary calcium. The coronary calcium level did not show any relationship with sitosterol or with campesterol. Otherwise, a clear association was found with plasma C. Windler et al. [65] published the results of the CORA study, in which 186 pre- and postmenopausal women with incident coronary heart disease were compared with 231 age-matched controls. The controls had significantly higher plasma concentrations of the major phytosterol species, but the cases had a higher dietary intake of phytosterols. No association was found between the plasma phytosterol concentration and coronary heart disease. In another recent study, Windler et al. [66] argue that homozygous patients with phytosterolemia exert extremely high (4000% increased) plasma phytosterol concentrations but a variable incidence of atherosclerosis. A coinciding elevation of plasma C obscures the conclusion that high plasma phytosterol concentrations may be responsible for atherosclerosis development. The authors refer to a meta-analysis of 41 randomized controlled trials with 55 treatment groups in a total of 2084 participants that showed that an average PS intake of 1.6 g/day in the form of fortified foods increases plasma concentrations of sitosterol and campesterol by on average 31% and 37%, respectively [67]. This small increase in plasma sitosterol and campesterol is in contrast to the results of Weingärtner et al. [58], who found increases of 300%. Of great interest is a recent publication by Scholz et al. [68]. The authors studied the relationship between phytosterols and coronary artery disease by performing a genome-wide meta-analysis of 32 phytosterol traits reflecting absorption, synthesis, and esterification in six studies with 9758 subjects. They detected ten independent genome-wide significant SNPs at seven genomic loci. A positive causal association was found between plasma sitosterol concentration and coronary artery disease. Part of this association was mediated by C. Table 1 shows studies with positive association, and Table 2 shows studies with no association between plasma phytosterol concentration and CHD risk or events.

## 6. Discussion

The discussion on the cardiovascular risk of phytosterols consumed in dietary supplements, in order to reduce plasma LDL-C, will be continued. The first principal question is whether the risk reduction obtained with the ~10% plasma LDL-C reduction exceeds the potential risk enhancement caused by the increased plasma phytosterol concentration. Literature data today do not answer this important question. A number of other aspects may be discussed here that have not been part of the previous discussion yet.

### 6.1. Natural Elimination of Phytosterols from the Human Body

In Section 3, it is indicated that the average fractional absorption rates for C, phytosterols, and -stanols are 50%, 10%, and near 1%, respectively. These numbers are established by the selective interaction of the different sterols with NPC1L1, sterol esterases, and ABCG5/G8 in the enterocyte. The interaction of phytosterols is low for NPC1L1 and esterase but high for ABCG5/G8. NPC1L1 and ABCG5/G8 are also present in the liver. Biliary-secreted C and phytosterols are partially absorbed back into the liver by NPC1L1. ABCG5/G8 stimulates biliary secretion. Also in the liver, phytosterols are only little re-absorbed and highly efficiently secreted into bile. This must be interpreted as a natural process to eliminate phytosterols from the body. Phytosterols appear to be harmful for humans, but why? In the patients with phytosterolemia, plasma phytosterol levels are extremely elevated compared to C, but extremely high degrees of atherosclerosis are not generally observed. Interestingly, increasing evidence is presented for potential benefits of phytosterols in many diseases like cancer and diabetes [75]. However, the question of why phytosterols are so badly absorbed and so strongly removed has been ignored so far.

### 6.2. Phytosterols and Atherosclerotic Risks in Phytosterolemia

Phytosterolemia or sitosterolemia is solely associated with ABCG5/G8 mutations [76]. A mutation in either one of the two genes encoding for ABCG5 ofABCG8, leading to reduced resecretion of sterols from the enterocyte back into the intestinal lumen and reducing biliary sterol secretion, affects both phytosterols and C (see Figure 2). However, the C resecretion rate is physiologically smaller and is affected to a lower degree by mutations too. The development of atherosclerosis may be ascribed to both phytosterols and C. Development of atherosclerosis in phytosterolemia patients is highly diverse. Interestingly, it has been observed that patients that established elevated plasma C in childhood develop atherosclerosis at a later age [77]. In children with homozygous phytosterolemia, extremely high C levels have been observed in the range similar to those with severe homozygous FH, which can lead to fatal myocardial infarction as early as five years of age [78,79]. Therefore, phytosterolemia has also been referred to as pseudo-homozygous FH [80,81] and is perhaps mainly a pediatric disease [66]. The role of the physiological phytosterol intake in the development of phytosterolemia has not been studied. Even further studies are necessary to study the behavior of high plant sterol supplementation in different stages of hypercholesteromia.

### 6.3. Determination of Clinical Endpoints After Long-Term High-Phytosterol Intake—Possible or Impossible?

It must be argued that the majority of clinical studies feeding 2 to 3 g esterified phytosterols have been performed for decades already. More recent studies engaged genetic information in preselected mostly phytosterolemia patients. The principal problem in clinical phytosterol research is to study hard clinical endpoints, such as cardiovascular morbidity and mortality, after long-term intake of high doses of phytosterols. Subjects consuming foods enriched with phytosterols are not registered and not available for research purposes. Long-term intake studies in huge populations of well-defined hypercholesterolemic patients compared to an age-, sex-, and plasma-C-matched placebo-treated group are necessary but impossible to perform. They require 10 to 20 years of follow-up of the patients, diverse clinical tests detecting atherosclerosis, and complex laboratory tests measuring plasma C in diverse lipoproteins and plasma phytosterols and -stanols. Financing must be taken care of by independent non-industrial suppliers and supervised by governmental institutions. At this moment, we can only give reference to studies indicating small-scale evidence for an increased atherosclerotic risk based on animal experiments and clinical metabolic and genetic studies.

### 6.4. Which Patients Should Be Treated with Plant Sterol Supplementation?

The purpose of phytosterol treatment is to reduce the risk of cardiovascular disease development by reducing plasma LDL-C [82]. The patient’s risk score must be established, i.e., being low, moderate, or high. Depending on the expected risk, the treatment must be adequately designed. A 10% reduction obtained with phytosterol treatment may make sense in patients at low risk. Treatment may extend the life span at low risk. Today’s situation is that the dietary supplements fortified with phytosterols are sold in local supermarkets. They are available for all customers visiting the supermarket. It may be expected that the majority of customers buying the dietary supplements have no knowledge about their plasma LDL value or their risk for cardiovascular disease. They expect to improve their health. Subjects with a normal plasma LDL-C will not benefit from the treatment, while the LDL-C-lowering efficiency is positively associated with the baseline LDL-C before treatment [48]. Otherwise, they are subjected to the potential risk of atherosclerosis development. This risk may be low, but dependent on the dosage, NPC1L1, and ABCG5/G8 polymorphisms. In particular, in variants with low ABCG5/G8 activity, the risk may be individually increased. Thus, phytosterol-enriched dietary supplements should be available for low-risk subjects with moderately elevated plasma LDL-C. In order to reach these subjects, the products should be removed from the supermarket and be supplied by the pharmacy after adequate risk assessment.

German and European authorities released critical guidelines with respect to food supplementation with phytosterols and draw clear attention to food supplementation [83]. The Task Force for the management of dyslipidemias of the European Society of Cardiology (ESC) and European Atherosclerosis Society (EAS) established guidelines for the use of phytosterols in order to lower LDL-C in 2021 [43]. The following was stated: “Daily consumption of 2 g of phytosterols can effectively lower Total C and LDL-C levels by 7 to 10% in humans (with a certain degree of heterogeneity among individuals), while it has little or no effect on high density lipoprotein C (HDL-C) and triglyceride (TG) levels. However, to date, no studies have been performed on the subsequent effect on cardiovascular disease (CVD). Based on LDL-C lowering and the absence of adverse signals, functional foods with plant sterols/stanols (≥2 g/day with the main meal) may be considered: (i) in individuals with high C levels at intermediate or low global CV risk who do not qualify for pharmacotherapy; (ii) as an adjunct to pharmacological therapy in high- and very-high-risk patients who fail to achieve LDL-C goals on statins or could not be treated with statins; and (iii) in adults and children (aged > 6 years) with FH, in line with current guidance”. A statement for healthcare professionals was published by the Nutrition Committee of the Council on Nutrition, Physical Activity, and Metabolism of the American Heart Association in 2001 [84,85]. It was stated that “until long-term studies are performed to ensure the absence of adverse effects in all individuals ingesting plant sterol esters, these products should be reserved for adults requiring lowering of total and LDL-C levels because of hypercholesterolemia or the need for secondary prevention after an atherosclerotic event. Whether plant sterols should be used in normocholesterolemic individuals with other risk factors for coronary heart disease (e.g., low HDL-C levels) remains to be determined”. These guidelines indicate the necessity to apply phytosterols to patients with a defined level of hypercholesterolemia and established risk to develop cardiovascular disease.

In the LURIC study, Silbernagel and colleagues demonstrated that high C absorption and low synthesis are associated with coronary heart diseases and cardiovascular mortality [86]. It is well established that a high absorption rate for C is in line with a higher uptake of further xenosterols, such as campesterol and sitosterol [13,87]. Despite improved treatments of chronic kidney disease (CKD), patients are still affected by an inappropriate high cardiovascular morbidity and mortality [88]. In patients on hemodialysis, lowering of LDL-C with statins only did not show a reduction in cardiovascular events [89,90], while LDL-C lowering could successfully reduce major cardiovascular events in non-dialysis patients [91]. Within the SHARP study, patients with advanced CKD with and without dialysis showed a reduction in cardiovascular events when statins were combined with ezetimibe, which additionally lowers intestinal C absorption and plant sterols besides total C and LDL-C. Genser et al. performed a post hoc analysis in 1030 participants in the German Diabetes and Dialysis Study (4D) who were randomized to either 20 mg atorvastatin or placebo. The primary endpoint was a composite of major cardiovascular events. Tertiles of the cholestanol-to-C ratio were used to identify high and low C absorbers. Those with low C absorption appear to benefit from treatment with atorvastatin, whereas those with high absorption did not benefit [92]. Interestingly, patients characterized as “high-absorber” have worse clinical outcomes than high synthesizers. Thus, common “high-absorbers” for cC, plant sterols, and other xenosterols should absolutely avoid intake of high amounts of phytosterols, especially strictly avoiding phytosterol-enriched food supplements. However, how should they know about their absorber habits? For this, they have to perform individual tests, such as specialized plasma C and phytosterol analysis as proposed by our group [13]. We are curious if subjects, using phytosterol-supplemented compounds freely available in the supermarket, do know anything about their C and phytosterol uptake and secretion.

### 6.5. What Is the Value of Plasma Phytosterol Concentrations in the Assessment of Risk for Atherosclerosis?

In animal experiments, wild-type mice or mice with humanized C metabolism are divided in two groups, consuming the phytosterol-enriched diet or the control diet. After the treatment period, the animals are sacrificed, and blood is collected for sterol analysis. Does the plasma phytosterol concentration at the moment of blood collection reflect the burden of the ingested phytosterol load during the treatment period? In human experiments, humans are studied in two phases, consuming a phytosterol-enriched diet and a control diet with a wash out period in between. After the two treatment periods, fasting blood samples are collected for sterol analysis. In humans, phytosterol-fortified food is consumed once, twice, or three times daily, together with meals. The last meal is normally consumed in the early evening. Thus, the daily burden of phytosterols is during the day. So, what is the meaning of the fasting plasma concentration to reflect the risk for atherosclerosis development? Baumgartner et al. [93] studied the postprandial response of phytosterols and oxyphytosterols to a phytosterol-fortified breakfast. The results were unique, showing that the postprandial increase in plasma phytosterol concentration occurred very slowly and had to be stimulated by a second meal after four hours not enriched in phytosterols. After eight hours, the plasma concentration reached 0.03 and 0.02 mg/dL for campesterol and sitosterol, respectively. Normal mean values in the fasting plasma of healthy controls are 0.5 and 0.3 mg/dL for campesterol and sitosterol, respectively [31]. In a previous paper, the authors showed that after a 4 weeks period using the same meals, the fasting plasma ratio’s to C of sitosterol and campesterol were both elevated 60% [94], much less than the threefold increase described by Weingärtner at al [58]. A link to the postprandial data cannot be made. Consuming a similar amount of phytostanols instead of phytosterols, the fasting plasma ratio’s for sitostanol and campestanol both increased approximately 5-fold. Apparently, phytostanols accumulate more in plasma. Does that express a higher risk for atherosclerosis development? The reductions in plasma LDL-C were similar under both treatments being on average 8% with both sterols and stanols. What has the largest effect: the reduction in the atherosclerosis risk by the reduction in plasma LDL-C or the induction of atherosclerosis risk by the increased intake of phytosterols? It is amazing that the 10-fold-increased phytosterol intake and 100-fold-increased intake of phytostanol lead to only a 60% increase in the plasma phytosterol and a 5-fold increase in the plasma phytostanol. This proves that the protective mechanism eliminates phytosterols from the body.

### 6.6. On the Association Between High-Dose Phytosterol Intake and Development of Atherosclerosis

The causality of atherosclerosis development under phytosterol treatment is a major concern. Clinical studies always have a treatment period limited to some weeks maximally. The study of atherosclerosis development in subjects on long-term consumption of phytosterol-enriched foods bought in the supermarket is nearly impossible. Firstly, the required number of subjects to compare would be extremely high, and secondly, a treatment for 20 years as a minimum would be necessary. In a real-world study, the subjects would not be registered, may irregularly interrupt their intake, and will regularly change the dose. Intake will most likely not be continued for a longer period of time. Also, most subjects are neither aware of their basal cardiovascular risk nor of their plasma LDL-C concentration before starting consumption and the progress or success in total and LDL-C lowering in plasma periodically during the intake of the phytosterol-enriched food. In preclinical (wt and ApoE k.o. mice) and clinical investigations, Weingärtner and colleagues [58] showed that plant sterol supplementation impairs endothelial function (wild-type mice), aggravates ischemic brain injury (mice stroke model), affects atherogenesis in mice, and leads to increased tissue sterol concentrations (human aortic valve cusps). In another study in ApoE k.o. mice, Weingärtner et al. found significant differences in the use of plant sterol esters (PSEs) and plant stanol esters (PSAs). Atherosclerotic lesion retardation was more pronounced in WTD + PSA, coinciding with higher regenerative monocyte numbers, decreased oxidative stress, and decreased inflammatory cytokines compared with WTD and PSE [95]. These data may be considered mechanistic proofs of atherosclerotic effects of phytosterol treatment. Seventeen years ago, the urgent call for prospective clinical studies with cardiovascular end points for functional foods supplemented with PSE, advertised for patients with cardiovascular diseases, was already expressed [58]. Instead of performing such clear endpoint studies, nothing has happened, just the opposite; these supplementations included in the diet are currently claimed from industry-friendly and -sponsored publications as heart-healthy diets [96]. However, on the other hand, some positions have changed, and also, scientists, who so far recommended plant sterol supplementation, now expressed criticism of the use of plant sterols, stating that a Mendelian randomization analysis reveals that phytosterols are polygenic traits and supports strong evidence that “sitosterol have a direct and indirect causal effect on coronary artery disease” [68].

## 7. Conclusions

Currently, plasma LDL-C lowering by pharmaceutical means is well defined. The patient’s risk can be defined, and treatment can be adjusted accordingly. Neutraceutical therapy is mainly created by dietary supplements enriched with physterols or phytostanols or a combination of both. A dosage of 2 to 3 g a day is advised, which leads to an average LDL-C reduction of about 10%. This dosage expresses a 10-fold-increased intake for phytosterols or a 100-fold-increased intake of phytostanols. Phytosterol and -stanol treatment is provided by dietary supplements that are directly available in the supermarket. Scientific evidence has been presented that increased phytosterol intake may be associated with an increased risk for development of atherosclerosis. The degree of increased risk is dependent on the patient’s genetic polymorphisms in NPC1L1 and ABCG5/G8 transport proteins as well as the established risk reduction due to reduction in LDL-C. Subjects with a normal or only slightly elevated plasma LDL-C have no or only minimal LDL-C reduction and lack the compensation for the potential increased risk for atherosclerosis. For subjects with a highly elevated plasma LDL-C, the 10% reduction is insufficient. Plasma LDL-C reduction must be performed in patients with a well-documented risk score for atherosclerosis and a defined level of required LDL-C reduction.

Although phytosterol-fortified dietary supplements are considered natural food products, they serve as a medication to lower plasma LDL-C. Their use must be preceded by confirmation of an elevated plasma LDL-C concentration and by confirmation of additional risk factors for development of cardiovascular disease. Adequate LDL-C reduction has to be confirmed regularly and guided by medical care. At larger LDL-C elevations, pharmaceutical treatment with a statin with or without ezetimibe must be started. Patients on long-term treatment with phytosterols must be checked for their genetic polymorphisms in NPC1L1 and ABCG5/G8 transport proteins in order to avoid phytosterol aggregation and induction of atherosclerosis. These conditions are not in line with a free availability of the products to everybody without medical advice.

## Figures and Tables

**Figure 1 nutrients-17-00654-f001:**
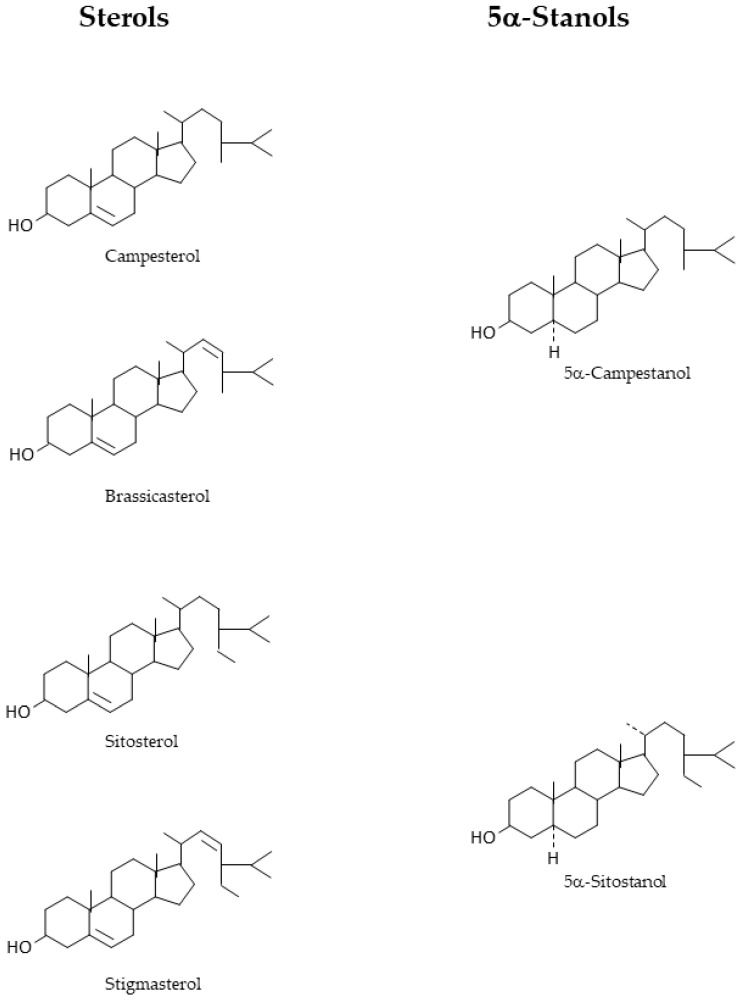
Chemical structures of the most prominent plant sterols and their corresponding 5α-stanols. (Created by Powerpoint 2016).

**Figure 2 nutrients-17-00654-f002:**
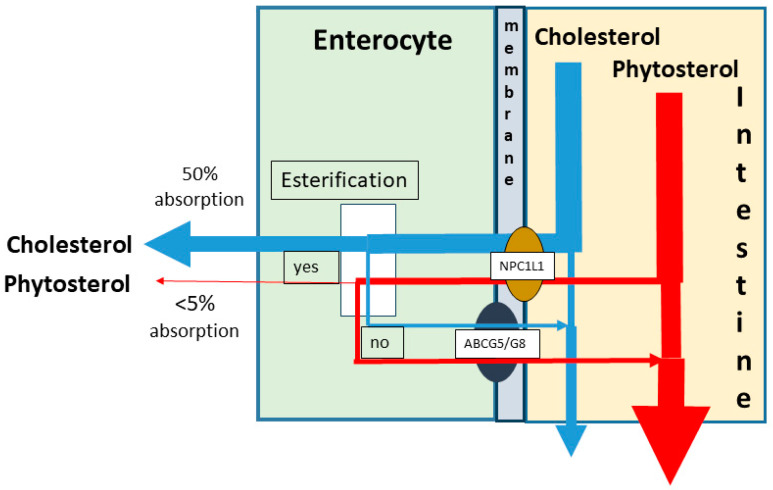
Schematic overview: Absorption of cholesterol and phytosterols. NPC1L1, Niemann–Pick C1-Like 1; ABCG5/G8, Adenosintriphosphate-binding-cassette tandem transporter G5/G8.

**Table 1 nutrients-17-00654-t001:** Studies showing a statistically significant association between plasma plant sterols and cardiovascular events/risks.

Subjects	Results	Comments	Study
Verified CAD (*n* = 48), controls (*n* = 61)	Plasma campesterol and sitosterol-to-cholesterol ratios were statistically significant associated with CAD (*p* < 0.05 for both). Odds ratios 1.01; 1.00 to 1.01 and 1.01; 1.00 to 1.03, respectively.	Statin intake not reported. Postmenopausal women only.	Rajaratnam et al., 2000 [69]
Patients with (*n* = 26) or without (*n* = 27) CHD family history	FH patients had higher plasma ratios of campesterol and sitosterol to cholesterol (*p* = 0.006 and *p* = 0.004, respectively)	Lack of a true control group. Statin intake and dietary intake of plant sterols not reported.	Sudhop et al., 2002 [70]
Cohort study, cases with coronary events (*n* = 159), controls (*n* = 318)	Cases had elevated absolute plasma sitosterol concentrations (*p* = 0.030).	CHD risk factors, i.e., LDL-cholesterol not matched between cases and controls. Conclusions drawn based on sitosterol only. Uni-variate analysis only.	Assmann et al., 2006 [71]

**Table 2 nutrients-17-00654-t002:** Studies showing no association between plasma plant sterols and cardiovascular events/risks.

**Subjects**	**Results**	**Comments**	**Study**
Hypercholesterolemic subjects. 231/364 (m/f)	Cholesterol correlated weakly with plasma campesterol and sitosterol. High campesterol was tendentially but not statistically significant associated with family history of CHD.	Exclusion criteria and statin intake not reported.	Glueck et al., 1991 [72]
People with family history of CHD 413/619 (m/f) People without family history of coronary heart disease (CHD) 807/619 (m/f)	Family history for CHD is not associated with elevated absolute plant sterol levels and plant sterol-to-cholesterol ratios. Plasma sitosterol unrelated to artery calcium score.	Large sample size. Age of subjects younger than in other studies. No absolute plasma plant sterol concentrations reported.	Wilund et al., 2004 [64]
Nested control study Cases with coronary events; 232/141 (m/f) Controls *n* = 758	Absolute campesterol and sitosterol concentrations were not different between cases and controls. Sitosterol-to-cholesterol ratio was lower in cases than controls. (*p* = 0.008) Campesterol-to-cholesterol ratio not different.	Large sample size. Adjustent for major risk factors established by multivariate analysis.	Pinedo et al., 2007 Epic-Norfolk cohort [73]
Community based cross-sectional *n* = 1192; 47% male *n* = 125 with CHD	Plasma plant sterols and their ratios to cholesterol slightly but significantly lower in subjects with CHD compared to subjects without CHD. High plasma sitosterol concentrations associated with a markedly reduced CHD risk (Odds ratio (OR) 0.78;95 CI 0.62–0.98.	Sitosterol higher in females than in males. Sitosterol lower in diabetics than in non-diabetics.	Fassbender et al., 2008 Longitudinal Aging Study Amsterdam (LASA) [74]

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
