# Peer review of "Phytosterol-Enriched Dietary Supplements for Lowering Plasma LDL-Cholesterol: Yes or No?"

_nutrients, 2025, doi:10.3390/nu17040654_

Round 1

Reviewer 1 Report

Comments and Suggestions for Authors

The manuscript provides a review of the use of phytosterols in reducing serum LDL-c levels, which can decrease serum LDL-c by 10%, but may be associated with an increased risk of atherosclerosis. Several issues are needed to be addressed.

Major comments

1.      I am curious about how low bioavailability of phytosterol can be a risk for human or animals? If so, the phytosterol in gut lumen but not the absorbed parts may be the main way for its biological function.

2.      Several descriptions are less of strong and newly references. Specifically, the relationship results from clinical data are not the direct causality for phytosterol and atherosclerosis.

3.      More direct evidences are needed for supporting the conclusion. For instance, how about the role of high level of phytosterol from cereal in human health and diseases?

Minor comments

1. In the abstract, the first occurrence of an abbreviation should be accompanied by its full name, for example, Cholesterol (C).

2. In Figure 2, the position of NPC1L1 and ABCG5/G8 within the epithelial cells is depicted, with esterified cholesterol being absorbed and unesterified cholesterol being effluxed. By modifying the position of ABCG5/G8 in the figure, the role of ABCG5 in the absorption and efflux of cholesterol and phytosterols can be more intuitively represented.

4. This article describes and compares the process of cholesterol absorption and phytosterol absorption; however, there is limited description of the cholesterol-lowering function of substances such as phytosterols. It would be appropriate to supplement this content accordingly.

5. In the section on Pharmaceutical Treatment, the sentence describing the dosage of ezetimibe, This dose is generally well tolerated, although many common but tolerable side effects may occur in less than 4% of the patients, lacks a reference.

6. In the section on Phytosterol Treatment, references 36, 38, and 39 are cited with formatting errors.

7. The manuscript is deficient in providing examples of adverse effects resulting from excessive phytosterol intake in humans or animals. Is phytosterolemia the text solely associated with ABCG5/G8 mutations? The relationship between phytosterolemia, atherosclerosis, and the intake of plant sterols is not thoroughly discussed. Please reevaluate and organize the evidence accordingly.

8. In the discussion section of the manuscript, the evidence base appears to be weak. It is recommended to enhance the support for your arguments by incorporating additional and up-to-date references. 

9. In Phytosterol application section, author presents that “Assuming that the higher intake of phytosterols and stanols does not lead to alteration of their fractional absorption rate, their daily absorbed amounts (mg/d) also increase 10 fold and 100 fold. ” while in the discussion section “It is amazing that the 10 fold increased phytosterol intake and 100 fold increased intake of phytostanol leads to only 60% increase in plasma phytosterol and 5 fold increase in plasma phytostanol.” Please explain the reasons for the inconsistent descriptions and list the relevant references.

Author Response

Answers to Reviewer 1 Comments and Suggestions for Authors

Rev #1: The manuscript provides a review of the use of phytosterols in reducing serum LDL-c levels, which can decrease serum LDL-c by 10%, but may be associated with an increased risk of atherosclerosis. Several issues are needed to be addressed.

Authors:

Common answer:

We are grateful to the reviewer’s comments, criticism, and questions.

The major comments of reviewer #1 are focused on the causality of phytosterols to establish an increased risk for development of atherosclerosis. We like to make a general statement to this topic: Nearly all review articles within the last 25 years end up in the conclusion as given in a recent review by Li X et al. 2022 [Li X, Molecules; 27:524. https://doi.org/ 10.3390/molecules27020523]: “However, the cholesterol-regulating effect remains to be further investigated. In addition, there is controversy as to whether PS has a pro-atherogenic effect in vivo, and extensive experimental confirmation is needed.” … “More, larger population-based multicenter studies, combined with animal studies are required to further demonstrate the efficacy and safety of PS, and shed light on its potential therapeutic and preventive roles and applications in clinical treatment and daily life.” This controversy asks to decide to promote the use of phytosterols based on the lack of definite proof of causality or to disapprove the use based on the lack of definite proof of non-causality. It is to our opinion irresponsible to offer nutritional supplements in such a free manner on the food market without the complete solution of the above given problems. Plant sterols in such high dosages such as 1.5 g – 3 g/day interfere significantly with the metabolic system in mammalians, are thus pharmacologically active agents and belong into the hands of medical-educated staff in order to avoid health injury for the user.

Rev. #1: Major comments:

Reviewer #1: 1. I am curious about how low bioavailability of phytosterol can be a risk for human or animals? If so, the phytosterol in gut lumen but not the absorbed parts may be the main way for its biological function.

Authors: The reviewer brings up the interesting question where phytosterols exert their biological function? This question may be extended to “ What is the physiological function of phytosterols at all?” No clear answer can be given to this question.

At first, we like to state that in this review we are exclusively interested in the inhibitory effect of plant sterol and plant stanol ester enriched supplements on cholesterol uptake in the intestine, resulting in reduction of plasma total and LDL-cholesterol in many but not all users together with the increase of plasma total plant sterols, dependent on the genetic background (NPC1L1 and ABCG5/G8) of the users, which, under long-term treatment, finallycould lead to pro-atherogenic effects in mammalian artery vessels or vessels and tissues in common.

Thus, we exclude such contributions and ideas as given in the excellent papers of Miszczuk E et al. [Pharmaceutical 2024; 17:1-17https://doi.org/10.3390/ph17050557], Jefrei E et al. [Br J Nutrition 2024; 131: 935-943] and Evtyugin D D et al. [Molecules 2023; 28: 6526. https://doi.org/10.3390/ molecules28186526] claiming beneficial pharmacological properties in relation to gastrointestinal tract including anti-inflammatory and hepatoprotective activity as well as the anti-cancer properties and the impact on the gut microbiome.

Manuscript: Introduction, page 2, 2nd para. “The specific biological function of phytosterols is unknown. Physiological beneficial associations have been indicated mentioned in relation to gastrointestinal tract including anti-inflammatory and hepatoprotective activity as well as the anti-cancer properties and the impact on the gut microbiome {Miszczuk, 2024 #150;Jefrei, 2024 #232;Evtyugin, 2023 #234}.”

Reviewer #1: 2. Several descriptions are less of strong and newly references. Specifically, the relationship results from clinical data are not the direct causality for phytosterol and atherosclerosis.

Authors: We assume that the reviewer asks for more references proving the causality of the relationship between phytosterols and atherosclerosis. We have reorganized the discussion. We set-up 6 different subchapters:

Manuscript (Discussion):

6.1         Natural elimination of phytosterols from the human body

6.2         Phytosterols and atherosclerotic risks in Phytosterolemia

6.3       Determination of clinical endpoints after long-term high-phytosterol intake – possible or impossible?

6.4. Which patients should strictly avoid plant sterol supplementation?

6.5. What is the value of plasma phytosterol concentrations in the assessment of risk for atherosclerosis?

6.6. On the causality of high-dose phytosterol intake on development of atherosclerosis

Regarding to reviewer #1 2nd question we answer to this in a new chapter 6.6.

Manuscript (Chapter 6.6): “The causality of atherosclerosis development under phytosterol treatment is a major concern. Clinical studies always have a treatment period limited to some weeks maximally. The study of  atherosclerosis development in subjects on long term consumption of phytosterol enriched foods bought in the supermarket, is nearly impossible. Firstly, the required  number of subjects to compare would be extremely high and secondly, a treatment for 20 years as a minimum would be necessary. In a real-world, open-field study, the subjects would not be registered, may irregularly interrupt their intake and will regularly change the dose. Intake will most likely not be continued for a longer period of time. Also, most subjects are neither aware of their basal cardiovascular risk nor of their plasma LDL-C concentration before starting consumption and the progress or success in  total and LDL-C lowering in plasma periodically during the intake of the phytosterol-enriched food. In preclinical (wt and ApoE k.o. mice) and clinical investigations Weingärtner and colleagues [Weingärtner O. et al. JACC. 2008; 51: 1553-1561 . doi:10.1016/j.jacc.2007.09.074] showed that plant sterol supplementation impairs endothelial function (wild-type mice), aggrevates ischemic brain injury (mice stroke model), effects atherogenesis in mice, and leads to increased tissue sterol concentrations (human aortic valve cusps). In another study in ApoE k.o. mice, Weingärtner et al. found significant differences in the use of plant sterol esters (PSEs) and plant stanol esters (PSAs). Atherosclerotic lesion retardation was more pronounced in WTD + PSA, coinciding with higher regenerative monocyte numbers, decreased oxidative stress, and decreased inflammatory cytokines compared with WTD and PSE [Weingärtner O et al. Cardiovasc Res. 2011; 90: 484-492]. These data, already mentioned in the original manuscript, may be considered mechanistic proofs of atherosclerotic effects of phytosterol treatment. Seventeen years ago, the urgent call for prospective clinical studies with cardiovascular end points for functional foods supplemented with PSE, advertised for patients with cardiovascular diseases, was already expressed [Weingärtner O. JACC. 2008; 51: 1553-1561 . doi:10.1016/j.jacc.2007.09.074]. Instead of performing such clear endpoint studies, nothing has happened, just the opposite; these supplementations are nowadays claimed from industry-friendly and -sponsored publications as heart-healthy supplementations [Simonen P,Lipids Health Dis. 2024 Oct 21;23(1):341]. However, on the other hand some positions have changed and also scientists, who so far recommended plant sterol supplementation expressed criticism in use of plant sterols stating that mendelian randomization analysis reveals that phytosterols are polygenic traits and supports strong evidence that sitosterol have a direct and indirect causal effect on coronary artery disease [Scholz M. et al. Nat Commun. 2022; 10;13(1):143. doi: 10.1038/s41467-021-27706-6].”

Reviewer #1: 3. More direct evidences are needed for supporting the conclusion. For instance, how about the role of high level of phytosterol from cereal in human health and diseases?

Authors: The average intake of total phytosterols in a western diet is around 300 mg/day, comparable with the intake of cholesterol. A higher intake of plant sterols is expected in vegetarians and in particular in vegans. Their intake is nearly twice as high (Jaceldo-Siegl et al. Molecular Nutrition 2017; 61, https://doi.org/10.1002/ mnfr.201600828). As indicated in the reviewer’s item #1, under physiological intake conditions, phytosterols are not a health-risk, and may even be protective against atherosclerosis [Wang, Am J Clin Nutr . 2024, 119, 344-353]. However, a two-fold increase in intake cannot be compared to the 10 fold and 100 fold increased intakes when consuming foods enriched in phytosterols and phytostanols, respectively. Our conclusion is not based only on the potential atherosclerotic risk. The mean 10% plasma LDL-cholesterol reduction reached by phytosterol supplementation is too low in patients with strongly elevated LDL-cholesterol. In subjects with normal LDL-cholesterol, no reduction is possible. Thus, the target group of eligible patients under the consumers is small. But the consumer is unaware of his LDL-cholesterol. Also, the genetic mutation of ABCG5/G8 of the subject and therewith the atherosclerotic risk is not known. The treatment has a benefit only when the reduction of atherosclerotic risk due to the LDL-cholesterol reduction is larger than the induction of atherosclerotic risk by the phytosterols. In none of the consumers of phytosterol enriched foods, the benefit can be predicted.

 Minor comments:

Reviewer #1: 1. In the abstract, the first occurrence of an abbreviation should be accompanied by its full name, for example, Cholesterol (C).

Authors: We are grateful to the reviewer´s hint. We changed the first sentence accordingly.

Manuscript (Abstract 1. Sent.): “Elevated plasma low density lipoprotein cholesterol (LDL-C) is associated ...”

Reviewer #1: 2. In Figure 2, the position of NPC1L1 and ABCG5/G8 within the epithelial cells is depicted, with esterified cholesterol being absorbed and unesterified cholesterol being effluxed. By modifying the position of ABCG5/G8 in the figure, the role of ABCG5 in the absorption and efflux of cholesterol and phytosterols can be more intuitively represented.

Authors: Figure 2 has now been updated by indicating the positions more realistically.

Reviewer #1: 3.: This article describes and compares the process of cholesterol absorption and phytosterol absorption; however, there is limited description of the cholesterol-lowering function of substances such as phytosterols. It would be appropriate to supplement this content accordingly

In our original manuscript we mentioned the different working mechanisms of phytosterols and ezetimibe to establish reduction of the cholesterol absorption (original manuscript page 2, para 1). Phytosterols reduce competitively the uptake of cholesterol into the micelles in the proximal intestine before the processes of phytosterol and cholesterol absorption as intensively and firstly described by ikeda  et al. (Ikeda I. et al.1988 J Lipid Res; 29;1573-1582). Ezetimibe reduces absorption of phytosterols and cholesterol  by suppressing the activity of NPC1L1 (Altman SW Science 2001. 303:1201-1204).

Reviewer #1: 4.: In the section on Pharmaceutical Treatment, the sentence describing the dosage of ezetimibe, “This dose is generally well tolerated, although many common but tolerable side effects may occur in less than 4% of the patients,” lacks a reference.

Authors: The references “Bays HE, Clinical therapy 2001, 23:1209-1230 PMID: 11558859”  and “Shrank W et al. Med. Care 2012; 50:479-484. doi: 10.1097/MLR.0b013e31825517b6. is now included.

Manuscript (5.1. Pharmaceutical treatments, last sentence): “This dose is generally well tolerated, although many common but tolerable side effects may occur in less than 4% of the patients {Bays, 2001 #235;Shrank, 2012 #238}. “

Reviewer #1: 5.:In the section on Phytosterol Treatment, references 36, 38, and 39 are cited with formatting errors.

We carefully formatted textual hints to references and the list of references.

Reviewer #1: 6.:The manuscript is deficient in providing examples of adverse effects resulting from excessive phytosterol intake in humans or animals. Is phytosterolemia the text solely associated with ABCG5/G8 mutations? The relationship between phytosterolemia, atherosclerosis, and the intake of plant sterols is not thoroughly discussed. Please reevaluate and organize the evidence accordingly.

Authors: We are very grateful to this suggestion. We have inserted a new small chapter in Discussion entitled 6.1. Phytosterols and atherosclerotic risks in Phytosterolemia.

Manuscript (Discussion): “6.2 Phytosterols and atherosclerotic risks in Phytosterolemia

Phytosterolemia or sitosterolemia is solely associated with ABCG5/G8 mutations {Fong, 2021 #249}. A mutation in ABCG5/G8 leading to reduced resecretion of sterols from the enterocyte back into the intestinal lumen and reducing biliary sterol secretion, affects both phytosterols and cholesterol (see Fig. 2). However, the cholesterol resecretion rate is physiologically smaller and is affected to a lower degree by mutations, too. The development of atherosclerosis may be ascribed to both phytosterols and cholesterol. Development of atherosclerosis in phytosterolemia patients is highly diverse. Interestingly, it has been observed that patients that established elevated plasma cholesterol in childhood, develop atherosclerosis at later age {Park, 2014 #255}. In children with homozygous phytosterolemia, extremely high cholesterol levels have been observed in the range similar to those with severe homozygous FH, which can lead to fatal myocardial infarction as early as five years of age {Berge, 2000 #241;Kidambi, 2008 #243}. Therefore, phytosterolemia has also been referred to as pseudo- homozygous FH {Morganroth, 1974 #247;Yoshida, 2000 #245} and is perhaps mainly a pediatric disease {Windler, 2023 #177}. The role of the physiological phytosterol intake in the development of phytosterolemia has not been studied. Even more further studies are necessary to study the behaviour of high plant sterol supplementation in different stages of hypercholesteromia.”

Reviewer #1: 7.:In the discussion section of the manuscript, the evidence base appears to be weak. It is recommended to enhance the support for your arguments by incorporating additional and up-to-date references. 

Authors: we added a further para in the discussion entitled: 6.3 Determination of clinical endpoints after long-term high-phytosterol intake – possible or impossible?

Manuscript: “6.3 Determination of clinical endpoints after long-term high-phytosterol intake – possible or impossible?

It must be argued that the majority of clinical studies feeding 2 to 3 grams phytosterols and/or phytostanols have been performed for decades already. More recent studies engaged genetic information in preselected mostly phytosterolemia patients. The principal problem in clinical phytosterol research is to study hard clinical endpoints such as cardiovascular morbidity and mortality after long-term intake of high doses of phytosterols. Subjects consuming foods enriched with phytosterols are not registered and not available for research purposes. Long term intake studies in huge populations of well-defined hypercholesterolemic patients compared to an age, sex and plasma cholesterol matched placebo treated group are necessary but impossible to perform. They require 10 to 20 years of follow up of the patients, diverse clinical tests detecting atherosclerosis and complex laboratory tests measuring plasma cholesterol in diverse lipoproteins and plasma phytosterols and -stanols. Financing must be taken care of by independent non-industrial suppliers and supervised by governmental institutions. At this moment we can only give reference to studies indicating small scale evidence for an increased atherosclerotic risk based on animal experiments and clinical metabolic and genetic studies.

Reviewer #1: 8.:In Phytosterol application section, author presents that “Assuming that the higher intake of phytosterols and stanols does not lead to alteration of their fractional absorption rate, their daily absorbed amounts (mg/d) also increase 10 fold and 100 fold. ” while in the discussion section “It is amazing that the 10 fold increased phytosterol intake and 100 fold increased intake of phytostanol leads to only 60% increase in plasma phytosterol and 5.fold increase in plasma phytostanol.” Please explain the reasons for the inconsistent descriptions and list the relevant references.

Authors: As mentioned in the original manuscript (page 8, section 4.2. Phytosterol application) , assuming unaltered fractional absorption, the body exposure increases 10- and 100- fold, respectively, too. This enhances the potential risk of atherosclerosis development due to increased plasma phytosterol levels.

So far, no data have been published on the fractional absorption rates of phytosterols and phytostanols under extreme feeding conditions creating a 10-fold to100-fold increased daily intake. Attempts from our lab using stable isotope labelling failed, because of the overlap of M+ signals of the huge amounts of campesterol/sitosterol and campestanol/sitostanol esters which made isotope absorption studies using deuterium labelled sterols or stanols impossible. As indicated in our original manuscript (page 10, section 6.3), the role of plasma phytosterol concentration as a reflection of atherosclerotic risk is unclear. The “60% increase in plasma phytosterol and 5-fold increase in plasma phytostanol” is a single observation that has been measured in fasting plasma after a certain treatment period [Baumgartner, S Atherosclerosis 2013, 227, 414-419, doi:10.1016/j.atherosclerosis.2013.01.012.]. The absolute fasting plasma phytosterol concentrations and the elevations during a feeding experiment differ largely between publications. Only limited publications are available due the limited availability of analytical techniques to measure the extremely low plasma phytosterol and even lower phytostanol concentrations. Postprandial elevation of the plasma concentration after a single dose is surprisingly low and dependent on a second meal effect (original manuscript, page 11, section 6.3, ref 58). Consuming the daily dose in one, two or three portions may be expected to result in different postprandial responses. This aspect of phytosterol research needs further clarification. Extensive stable isotope studies should show the time dependent appearance of labeled phytosterols in plasma and the fecal excretion under basal conditions and under feeding conditions. 

Reviewer 2 Report

Comments and Suggestions for Authors

The authors presented an interesting article ‘Phytosterol enriched food supplements for plasma LDL-choles-terol lowering: Yes or no?’ in which they discussed the issue of the efficacy of phytosterols in lowering plasma LDL-cholesterol. The article is well-written and the topic addressed by the authors is timely. To improve the quality of the manuscript:

1. I suggest separating the ‘Materials and methods’ section where the authors should include the defined research question, information on searching medical information sources, and inclusion and exclusion criteria. This will allow the potential reader to understand by what criteria the authors selected the articles for review. This is not clear in this form.

2. Figure 1. Please include information on the software used to prepare the chemical formulae.

3. Figure 2. There are abbreviations in the figure which should be explained in the figure caption.

4. Page 8. The authors write: ‘In human experiments,...’ without stating which studies they are referring to, and there are no literature references.

5. I feel that the verbatim quotations from other sources in the manuscript (page 8) distort the perception of the content.

6. The authors should add the statistical significance of the results of the studies included in the review.

7. Because of the information in the text of the article on the side effects of statins, I suggest adding the recommended strategies for managing suspected intolerance.

8. The authors should clarify the differences between functional foods, dietary supplements, and nutraceuticals.

Author Response

Answers to Reviewer 2 Comments and Suggestions for Authors

Rev #2: The authors presented an interesting article ‘Phytosterol enriched food supplements for plasma LDL-choles-terol lowering: Yes or no?’ in which they discussed the issue of the efficacy of phytosterols in lowering plasma LDL-cholesterol. The article is well-written and the topic addressed by the authors is timely. To improve the quality of the manuscript:

Rev #2. 1. I suggest separating the ‘Materials and methods’ section where the authors should include the defined research question, information on searching medical information sources, and inclusion and exclusion criteria. This will allow the potential reader to understand by what criteria the authors selected the articles for review. This is not clear in this form.

Authors: Our paper is meant to be a review article, not a meta-analysis on various studies including thousands of patients. The literature availability is limited and the clinical studies performed in small study populations. Phytosterol treatment has been studied in preselected probands being healthy subjects or patients with various levels of basal plasma LDL-C. The majority of studies describe the plasma LDL-C reduction under intake of phytosterols. A smaller proportion deals with phytosterol metabolism itself. An even smaller group indicates evidence for the potential atherogenic risk of phytosterol feeding. The purpose of our review was to picture the scientific papers promoting or disapproving the use of phytosterol food supplements. 

Rev #2. 2. Figure 1. Please include information on the software used to prepare the chemical formulae.

Authors: We added this information in the legend for Fig. 1

Manuscript: Figure 1. . Chemical structures of the most prominent plant sterols and their corresponding 5a-stanols. (Created by Powerpoint 2016)

Rev #2. 3. Figure 2. There are abbreviations in the figure which should be explained in the figure caption.

Authors: Abbreviations have now been explained in the figure caption of figure 2.

Manuscript: Figure 2. Schematic overview: Absorption of cholesterol and phytosterols. NPC1L1, Niemann-Pick C1-Like 1; ABCG5/G8, Adenosintriphosphate-binding-cassette tandem transporter G5/G8.

Rev #2. 4. Page 8. The authors write: ‘In human experiments,...’ without stating which studies they are referring to, and there are no literature references.

Authors: We tried to simplify the statement by mentioning the generally used double blind, placebo controlled protocol used in most human clinical studies. This includes one study group being treated with an agent or a placebo in two regimens separated by a wash out phase. This protocol is independent of phytosterol research.

Rev #2. 5. I feel that the verbatim quotations from other sources in the manuscript (page 8) distort the perception of the content.

Authors: It is unclear to us what the reviewer means exactly. Does he mean that the quotations in the tables 1 and 2 should be removed? Why?

Rev #2. 6.: The authors should add the statistical significance of the results of the studies included in the review.

Authors: The reviewer’s request would be valid when we intended to write a meta-analysis paper trying to compare the numbers papers with positive and negative results on a selected research question. We further refer to the answer to the reviewer’s topic one.

Rev #2. 7.: Because of the information in the text of the article on the side effects of statins, I suggest adding the recommended strategies for managing suspected intolerance.

Authors: In our original manuscript we already mentioned that a statin can be replaced by bempedoic acid when statin intolerance is observed. We added that intolerance to a certain dose of a statin will be followed by the choice of another statin. Depending on the risk score of the patient, the dose can be reduced and / or ezetimibe added.

Manuscript: (Introduction, page 2, sent. 6): Pharmacological treatment is strongly advised in the responding guidelines for treatment of hypercholesteromia, which starts with statin treatment, that reduces endogenous C synthesis and increases LDL-receptor (LDL-R) activity [Parhofer KG. Dtsch Arztebl Int. 2016;113:261-268].

(Introduction, page 2, sent. 9.): Intolerance to a statin will be followed by the choice of another statin.

(Introduction, page 2, sent. 16.) In case that high-dose statin treatment is not well tolerated or in case that the LDL-cholesterol lowering by a statin does not reach the intended goal for LDL-lowering, statin dose can be lowered and treatment is combined with ezetimibe [Parhofer KG. Dtsch Arztebl Int. 2016;113:261-268].

Rev #2. 8.: The authors should clarify the differences between functional foods, dietary supplements, and nutraceuticals.

Authors: The phytosterol enriched food products are generally called dietary supplements. Is this expression correct? It implies that a dietary component is added to food in order to enhance its daily intake. This is the case for phytosterol enriched foods. The American food and drug administration (FDA) describes dietary supplements as “products intended to supplement the diet that bear or contain one or more of a number of dietary ingredients”. One of the listed ingredients  is described as “ a dietary substance for use by man to supplement the diet by increasing the total dietary intake. [https://www.fda.gov/food/dietary-supplements-guidance-documents-regulatory-information/dietary-supplement-labelinh-chapter-i-general-dietary-supplement-labeling#1-1]. Thus,

 we take over the term dietary supplement and keep this in line within the whole manuscript.

Round 2

Reviewer 1 Report

Comments and Suggestions for Authors

I have reservations about the conclusion. Although the authors replied point by point, there might be no direct evidences to support their viewpoint.

Author Response

Please find the response in the attachment

Reviewer 2 Report

Comments and Suggestions for Authors

I appreciate the corrections made by the authors. However, from the point of view of the potential reader, it would be advisable to include in the manuscript an explanation of what guided the authors in selecting publications from the period 1953-2024. Given the dietary supplements, nutraceuticals and functional foods available on the market, it would be appropriate to define in the manuscript the dietary supplements discussed and indicate the current regulations in this regard. Table 1 is very well drafted. However, I do not see the relevance of quoting other authors directly ("genetic variation in cholesterol absorption affects levels of circulating non-HDL choles-terol and risk of CAD. Our results indicate that both dietary cholesterol and phytosterols contribute directly to atherogenesis”; "the findings of this study underline the need for prospective clinical studies with cardiovas-cular end points for functional foods supplemented with phytosterols that are currently advertised for patients with cardiovascular diseases"). Providing statistical significance along with the results of the studies analyzed in the review would allow the reader to assess their reliability. Sometimes articles are published in which the results of experimental studies have not been statistically analyzed. I absolutely do not suggest the authors to conduct a meta-analysis, because it is not the topic of the manuscript.

Author Response

(The authors gave the same response as above.)
